# Community Pharmacies’ Promotion of Smoking Cessation Support Services in Saudi Arabia: Examining Current Practice and Barriers

**DOI:** 10.3390/healthcare11131841

**Published:** 2023-06-25

**Authors:** Fahad Alzahrani, Yazeed Sandaqji, Abdullah Alharrah, Ramzi Alblowi, Samer Alrehaili, Waleed Mohammed-Saeid

**Affiliations:** 1Clinical and Hospital Pharmacy Department, College of Pharmacy, Taibah University, Madinah 42353, Saudi Arabia; 2College of Pharmacy, Taibah University, Madinah 42353, Saudi Arabia; yazeed.sandaqji@gmail.com (Y.S.); ramzialblowi@gmail.com (R.A.); sameralrehaili@gmail.com (S.A.); 3Department of Pharmaceutics and Pharmaceutical Technology, College of Pharmacy, Taibah University, Madinah 42353, Saudi Arabia; wneyaz@taibahu.edu.sa

**Keywords:** Saudi Arabi, community pharmacist, smoking cessation, tobacco control, confidence, barriers, current practices

## Abstract

Background: Community pharmacists’ interventions have been found to be highly effective. However, to date, there has been little information about pharmacists’ potential roles and perceived barriers to providing smoking cessation. This study aimed to assess the potential role of community pharmacists in supporting their patients to quit smoking by summarizing their self-reported level of current activities, confidence, and perceived barriers; Methods: A self-administered questionnaire was used for a cross-sectional study in Saudi Arabia. As part of the questionnaire, demographic characteristics were measured, as well as confidence, perceived berries, and level of smoking cessation activities (asking, advising, assessing, assisting, and arranging, including follow-up).; Results: 370 community pharmacists practicing responded to the survey. Pharmacists indicated high activity rates about advising and assessing patients in quitting smoking, with lower rates of assisting and arranging, including following up. The rate of recording smoking status was very low. There were significant differences between gender, source of education, years practicing as a licensed pharmacist, attended an education or training program on smoking cessation, and interest in providing smoking and perceived practice of smoking cessation. Pharmacists are confident about providing smoking cessation activities. Barriers to providing smoking cessation services include unable to follow up, lack of counseling space, lack of educational materials, and lack of time.; Conclusions: Saudi Arabian community pharmacists actively offer smoking cessation services, which may reduce smoking-related health issues. It appears that pharmacists need to be empowered by team-based, systematic, comprehensive approaches to overcome barriers and enhance their confidence.

## 1. Introduction

Tobacco smoke is one of the biggest causes of illness, disability, and death worldwide. The World Health Organization (WHO) reports Over 8,000,000 people die every year as a result of tobacco use, with more than 7,000,000 deaths directly related to tobacco use [1]. About 10 million individuals are estimated to die annually from tobacco-related diseases by 2030 if world consumption patterns remain the same [2]. Hence, smoking cessation with community pharmacists is a crucial strategy for decreasing preventable tobacco-related deaths [3].

According to the United States (U.S.) Department of Health and Human Services, tobacco smoking damages nearly every organ in the body, causing many fatal diseases such as cancer, cardiovascular diseases, chronic lung diseases, and stroke [4]. Previous research concluded that lifelong smokers have a high risk of atherosclerosis, chronic obstructive pulmonary disease (COPD), and myocardial infarction [5]. Tobacco smoking also increases the economic burden on families and healthcare systems [4].

Saudi Arabia has signed the WHO Framework Convention on Tobacco Control (FCTC), launched in 2003. It provides a framework for developing and implementing effective tobacco control policies, including advertising, promotion, and sponsorship measures to reduce tobacco consumption and its worldwide health effects [6]. In addition, smoking is prohibited in most places in Saudi Arabia, including healthcare facilities, educational institutions, government facilities, and places where the tobacco industry advertises and sponsors [7]. However, smoking consumption has increased significantly during the last decades [8,9]. Approximately 19.8% of adults smoke cigarettes [10]. This prevalence was reflected in 30.0% of males and 4.2% of females [11]. Further, smoking generates a cost of around 4545 million Saudi riyals in terms of its economic impact. This monetary burden includes direct and indirect costs associated with healthcare spending and decreased productivity because of smoking-related early mortality and morbidity [12]. Even though Saudi Arabia does not produce tobacco, it is among the top importers of tobacco products in the world [13]. The value of tobacco products imported between 2010 and 2014 was estimated at more than US $3.4 billion [14].

Past studies have proposed that smokers who quit smoking at any point can see a dramatic decline in mortality rates related to smoking-related cardiovascular diseases and lung diseases [15,16]. Consequently, exposure to anti-tobacco mass media campaigns, school programs, tax increases, and health messages on tobacco products can motivate smokers to quit [17,18]. However, without support and help, smokers experience difficulty discontinuing due to nicotine dependence [18]. For instance, a U.S. study found that most American daily smokers who attempt to quit without assistance will relapse to smoking [19].

Smoking cessation programs provide a solution to reducing the number of smoking-related health problems and deaths [20]. Smoking cessation has been proven to reduce the risks of cardiovascular disease, stroke, and cancer caused by tobacco at any age [21,22]. Smoking cessation is widely accepted as an effective clinical practice. Even a brief intervention by a healthcare professional increases the chance of quitting smoking significantly [23]. There is also growing evidence that smokers who receive a clinician’s assistance report better satisfaction with their healthcare than those who do not [23]. Prior research found that smoking cessation counseling encourages tobacco smokers to use a combination of pharmacological benefits and psychological assistance to stop this bad habit more efficiently [24]. The WHO reported that messages from healthcare professionals could help smokers stop smoking more efficiently [25].

Community pharmacists, with their accessibility and expertise in medication management, are well-positioned to play an essential role in smoking cessation [26,27,28]. Several systematic reviews have shown that community pharmacists’ involvement in smoking cessation intervention can lead to better quit rates and reduce the risk of smoking-related illnesses [29,30,31]. Community pharmacists could provide numerous pharmacological smoking cessation interventions, such as free and low-cost nicotine replacement therapy (e.g., NRT, such as gums, lozenges, and patches), that can be purchased from community pharmacies without a prescription. Moreover, they can advise and guide smokers after receiving proper and specific training or education on smoking cessation strategies [32].

Despite the potential benefits of including community pharmacists in smoking cessation efforts, their intervention in this area remains underutilized in Saudi Arabia. Therefore, this study aims to explore the confidence and the current involvement of Saudi Arabian community pharmacists in providing smoking cessation services and their perceived barriers to successful smoking cessation.

## 2. Materials and Methods

### 2.1. Study Design and Eligibility Criteria

The research involved 370 licensed community pharmacists in Saudi Arabia participating in a cross-sectional online questionnaire study between March and September 2022. A study population of Saudi Arabian community pharmacists with a bachelor’s degree or higher and at least 18 years of age were recruited to participate in this study. The inclusion of pharmacists was based on signing consent forms and responding to a questionnaire. All participants who failed to meet the inclusion criteria were excluded. In 2018, approximately 8419 pharmacists were employed in Saudi Arabian community pharmacies [33]. Using Raosfot^®^’s online sample size calculator, estimated that this study’s minimum adequate sample size was 368, with a 95% confidence interval and a 5% error margin.

### 2.2. Sampling Procedure

Community pharmacists were recruited through a variety of methods. A social media campaign was utilized to spread the word about the study among pharmacy professional groups and organizations. The study was also advertised through internal distribution lists at several national chain pharmacies. Other regional and rural pharmacies were also contacted directly to promote the survey for pharmacists outside urban areas. To avoid duplicate responses, participants were asked to complete the questionnaire only once, no matter how they received it.

### 2.3. Face Validity, Pilot Testing, Reliability, and Internal Consistency

A total of five researchers reviewed the questionnaire. Each item was rated on a Likert scale of 1–5 by three academic pharmacists and two community pharmacists with experience providing smoking cessation services. Items rated as not relevant by all researchers were excluded. Those items deemed relevant or highly relevant were retained. Discussion and consensus were used to resolve questionable items. Moreover, the study tool was piloted before it was used to ensure clarity and comprehensibility. In one round, the same pharmacy graduates (n = 10) completed the questionnaire, and after a short time interval (30 min to 1 h), they completed another copy. Score stability was tested using the test-retest method over a brief period of time. Using Pearson’s correlation, the scores for both rounds were compared. Based on the results of the pilot testing, the questionnaire survey in this study proved to have significantly stable scores over a short timeframe, as indicated by Pearson’s correlation coefficient of greater than 90% (95% CI = 0.90–0.98) with a *p*-value of <0.01. Based on previous research, acceptable coefficients were set a priori as >80% [34]. Cronbach’s alpha was utilized to test the internal consistency of the questionnaire items. To ensure internal consistency, Cronbach’s alpha should be above 70% [35]. The internal consistency of the items used in the test was excellent, as indicated by Cronbach’s Alpha of 94.6%. Cronbach’s alpha values were also computed for each domain separately. The Cronbach’s alpha was 93.2%, 95.5%, and 95.2% for eight items pertaining to confidence in undertaking different aspects of smoking cessation, 15 relating to items of roles and current practice, and 11 items representing barriers.

### 2.4. Study Tool

The questionnaire used for this study was designed and modified based on previously published studies with similar objectives [36,37,38]. The following measures were included:

#### 2.4.1. Background Variables

A variety of variables were measured, including personal characteristics (age, gender, level of education, source of education, smoking status) and practice characteristics (years practicing as a licensed pharmacist, work status, region of practice, type of community, attending smoking cessation education or training, interest in providing smoking cessation, average number of patients seen in the last week, average length of consultation per tobacco smoker).

#### 2.4.2. Confidence

Pharmacists’ confidence in undertaking smoking cessation activities was assessed using an eight-item scale. Harmacists’ confidence in undertaking smoking cessation activities was assessed using an eight-item scale such as “discussing readiness to change smoking behavior”.

#### 2.4.3. Smoking Cessation Activity

Fourteen scales addressed smoking cessation activity. Scales ranged from 1 “Never” to 5 “Always”. Recording smoking status was measured with two items, “Asking patients about their smoking cessation” and “Recording smoking status on file”. Three items measured advising: “Providing brief advice to quit”, “Linking advice to the presenting problem”, and “Discussing the effects of smoking on other family members”. Assessing was measured using two items, “Assessing interest in quitting” and “Assessing the level of nicotine dependence”. Assisting To bacco smokers to quit smoking was measured by four items: “Setting a quit date”, “Developing a cessation plan”, “Providing quit materials”, and “Recommending nicotine replacement therapies such as gum and patches”. Arranging was measured using two items: “Referring to the quitline program” and “Referring to other general practitioners or pharmacists”. In addition to the 5 As, follow-up activities were measured with one item: “Following up on progress in giving up smoking”.

#### 2.4.4. Barriers to Smoking Cessation Services

Seven items measured system barriers addressing situational constraints or systemic problems. Four items measured pharmacist barriers relating to personal factors. An example of this is the “Lack of skills to assist patients to quit”. A five-point Likert scale was used to assess the perceived impact of each barrier (1 being “No Impact” and 5 being “High Impact”).

### 2.5. Statistical Analysis

The data were analyzed using the Statistical Package for the Social Sciences (SPSS) (version 27 for Windows (SPSS Inc., Chicago, IL, USA). The study variables were summarized using descriptive statistics. A continuous variable is presented as a mean and standard deviation, while a categorical variable is presented as a frequency and percentage. Kolmogorov–Smirnov and Shapiro–Wilk tests were carried out to assess scores for normality of distribution. Without a normal distribution, The Mann–Whitney U and Kruskal–Wallis tests were utilized to evaluate the possible association between demographic and practice characteristics and smoking cessation activities. *p* < 0.05 and a 95% confidence interval (CI) indicated significant results.

### 2.6. Ethical Approval

The ethical research committee of the College of Pharmacy at Taibah University granted ethical clearance with the identification number (COPTU-RIC-30-20220222). Before enrolling in the study, all participants provided written informed consent and were briefed on the study’s purpose.

## 3. Results

### 3.1. Participant Demographics and Practice Characteristics

A total of 370 community pharmacists completed the online survey. The mean age of the community pharmacists was 32.1 ± 5.3 years; 37 (10.0%) were female in gender, 328 (88.6%) had a bachelor’s degree of Science (BSC) in pharmacy, 249 (67.3%) of the participants were practicing within the Western region, and only 79 (21.4%) of the community pharmacists graduated from Saudi universities. About 112 (30.2%) of the participants had an experience of fewer than five years, 237 (73.8%) were permanent employees, and 286 (77.3%) of the participants were practicing in a city. The majority of the participants (269, 80.0%) had never smoked cigarettes, and 251 (75.9%) of them stated that they were interested in providing smoking cessation services; however, only 71 (19.2%) had attended an educational or training program on smoking cessation. Almost more than half of the participants (189, 51.1%) had interacted with fewer than five cigarette smokers, and 263 (71.1%) stated that they spent, on average, five minutes or less with cigarette smokers. The detailed demographics and practice variables of the community pharmacists are shown in Table 1.

### 3.2. Smoking Cessation-Related Activities

For smoking cessation activities, 58 (15.6%) participants reported that they “always” or “most of the time” asked their patients about their smoking status. Only 25 (6.8%) of them reported that they always recorded the smoking status of their patients in a file, and 144 (38.9%) of the participants always or most of the time associated advice with the presenting visit. When the patients’ smoking status was identified, 152 (41%) of the participants responded that they always or most of the time provided brief advice to quit smoking, and 137 (37%) of the participants always or most of the time discussed the negative impacts of smoking on other family members. For patients willing to quit, 85 (23%) of participants responded that they always or most of the time were scheduled to stop. About 99 (26.8%) of the participants always or most of the time developed a cessation plan; 115 (31.1%) of the participants responded that they always or most of the time provided quit materials, and 129 (of the total 349) always or most of the time recommend the use of nicotine gum or patch.

With regard to referring cigarette smokers to a quitline program, 105 (28.4%) of the participants reported directing their patients to a smoking cessation program. Only 92 (24.8%) participants responded that they always referred their patients to other healthcare professionals. Finally, a quarter of the participants (93, 25.2%) answered that they always or most of the time followed up with their patients to quit smoking. Table 2 Summarizes the participants’ smoking cessation-related activities.

### 3.3. Association between Demographic and Smoking Cessation Current Activities

The Kruskal–Wallis H test was used to compare the mean differences in perceived practice scores among different groups of age, region of practice, level of education, the institution where the academic degree was obtained, number of practicing years as a pharmacist, type of community, smoking status, the average number of cigarette smoker interaction per week and an average length of consultation per cigarette smokers. In contrast, the Mann–Whitney U test was applied to compare the mean differences among gender, attendance at continuing education programs related to smoking cessation, and interest in providing smoking cessation services. The five demographic variables gender (*p* = 0.02), source of education (*p* = 0.01), years practicing as a pharmacist (*p* = 0.01), attendance at continuing education programs related to smoking cessation (*p* = 0.03), and interest in providing smoking cessation services (*p* = 0.01 were significantly associated with the current smoking cessation practice. On average, male pharmacists had higher smoking cessation activities than female pharmacists. Pharmacists who graduated from foreign universities had a higher mean score for smoking cessation activity than pharmacists from Saudi universities.

Furthermore, this study found that more experienced pharmacists scored higher than other categories in smoking cessation services. Pharmacists who attended continuing education programs related to smoking cessation had higher smoking cessation services mean scores than a pharmacist who did not participate in continuing education programs related to smoking cessation. Finally, the smoking cessation activities average mean score for pharmacists who were interested in providing smoking cessation services was higher than pharmacists who were not interested in offering smoking cessation services. On the other hand, other demographic variables were not significantly associated with the smoking cessation service mean score. The details of these associations are shown in Table 3.

### 3.4. Confidence and Perceived Barriers to Smoking Cessation-Related Activities

Community pharmacists’ confidence and barriers were analyzed to identify the specific issues perceived as the highest importance (as indicated by the mean responses for each item). The overall mean score of the level of participants’ confidence in performing the different aspects of smoking cessation counseling was 2.7 (67.5%) (1 = Not confident; 5 = Extremely confident). Overall, community pharmacists were confident about using straightforward advice to help patients quit smoking (M = 2.9, SD = 0.9), discussing patients’ readiness to change smoking behavior (M = 2.8, SD = 0.9), raising smoking issues when they are related to the pharmacy issues (M + 2.8, SD = 1.0), and increasing patient motivation to quit by using specific counseling strategies (M = 2.8, SD = 1.0). However, community pharmacists were least confident about raising smoking issues when not related to the pharmacy visit (M = 2.4, SD = 0.9) and engaging all staff members in a process to develop systems for smoking cessation (.M = 2.5, SD = 0.9).

Community pharmacists identified practitioner and system barriers as potential obstacles to providing smoking cessation services. Over half of the community pharmacists felt that each of the 12 potential barriers had some impact. The four most common barriers that community pharmacists cited impacting their ability to deliver smoking cessation services were related to system barriers. These barriers include the inability to follow up (M = 2.9. SD = 1.2), a lack of counseling space (M = 2.9, SD = 1.2), a lack of educational materials (M = 2.9, SD = 1.9), and a lack of time (M = 2.9, SD = 1.2). Regarding practitioner factors, the “patients might be alienated” barrier (M = 2.9, SD = 1.1) posed the most significant obstacle. However, community pharmacists did not perceive “not a worthwhile use of my time,”; “no plan to implement protocols/guidelines,” or “lack of necessary skills to assist patients in quitting” as having a high impact on their ability to provide smoking cessation services. The details of the responses are shown in Table 4.

## 4. Discussion

This study presents findings from a survey of community pharmacists concerning smoking cessation activities, confidence, and perceived barriers. To our best knowledge, it is the first Saudi Arabian study to:Examine the potential role that community pharmacists can play in helping smokers quit.Provide a comprehensive assessment of the barriers to smoking cessation activity among community pharmacists.Explore the relationship between community pharmacists’ smoking cessation activities and their demographic characteristics.

The study findings showed that a minority (21%) of community pharmacists in Saudi Arabia asked patients about their smoking habits every time or most of the time. As a result, many smoking patients remain unidentified. A previous study in Qatar showed that identifying patients’ smoking status is vital in providing smoking cessation services as it determines whether a tobacco user needs any interventions. The authors added that if pharmacists are waiting for patients to inquire about smoking cessation, they can only help patients ready to quit [38]. Thus, pharmacists should follow the 5 A’s—Ask, Advise, Assess, Assist, and Arrange, to assist more smokers in quitting. The Five A’s included asking patients about their smoking status; advising smokers to quit; assessing their willingness to stop smoking; assisting smokers in quitting; and arranging a follow-up with them [38,39]. Moreover, documenting patients’ smoking status encourages pharmacists to participate in smoking cessation activities [40]. Medical records of patients are currently not kept by Saudi community pharmacies. To ensure community pharmacists play an active role in smoking cessation counseling, Saudi community pharmacies should implement computerized software to track patients’ medical information.

Moreover, this study revealed that more than 35% of pharmacists assessed the interest of their patients in quitting when they identified them as smokers, offered brief advice on how to quit based on their understanding of the presenting issues, and discussed how family members were affected by smoking. These results indicate that community pharmacists have some role in preventing smoking-related harms. However, pharmacists can play a more significant role in smoking cessation activities, especially by assisting patients with setting a quit date, preparing a cessation plan, providing written educational materials, and arranging referrals and follow-ups. There are similarities between the current practice expressed by community pharmacists in this study and those described by the Australian study [36].

Currently, Saudi Arabian community pharmacies offer over-the-counter nicotine replacement therapies (NRT) for patients who wish to quit smoking. However, since there is no licensed E-cigarette, these products cannot be prescribed or provided, nor can they be recommended as a replacement for licensed medications [41].

Nevertheless, less than 35% of Saudi community pharmacists recommended NRT to tobacco smokers. As a result, Saudi community pharmacists ignore the opportunity to counsel tobacco users on quitting.

The study findings also revealed that more than half of the pharmacists (51.1%) had interacted with fewer than five tobacco smokers. A possible explanation for this might be the Ministry of Health in Saudi Arabia established anti-smoking clinics as a part of the National Tobacco Control Program (NTCP) to help those willing to quit smoking by providing them with free therapeutic services (behavioral and pharmacological) [42]. As a result, cigarette smokers prefer to visit these clinics over community pharmacies. Another possible explanation is that the public was unaware of advanced pharmaceutical services available from community pharmacists, such as smoking cessation, and untrusting pharmacists’ capability to provide such essential clinical services. Thus, patients did not turn to pharmacists for help quitting smoking. [43,44]. As this study shows, there is still much to be done to increase public awareness of community pharmacies’ clinical services, such as smoking cessation.

The present findings showed a significant association between community pharmacists’ practice of smoking cessation services and their gender, pharmacy educational sources, years of experience, prior attendance at education programs on smoking cessation, and interest in providing smoking cessation services.

Male pharmacists engaged more in smoking cessation services compared with female pharmacists. This may reflect the effect of Saudi socio-culture norms, which discourage female healthcare professionals from discussing or providing services to male patients. A recent study on Saudi female nurses in the Qassim region of Saudi Arabia reveals that more than 70% of nurses were unwilling to provide services to male patients. Most nurses also prefer to work in female units, and 51% of them did not like night shifts [45].

Additionally, those pharmacists who graduated from foreign universities engaged more in smoking cessation activities than those who graduated from Saudi universities. This finding may be that most Saudi Arabian pharmacy schools’ curriculums do not adequately prepare their graduate students to deliver clinical services, such as smoking cessation. A Saudi study discovered that most Saudi pharmacy schools do not train students in community pharmacy before graduation [46]. Moreover, pharmacy interns are only allowed to fill prescriptions in a pharmacy without any counseling or other service duties [47].

A significant association between years of experience with practices, where practices increased with an increase in the years of working experience. The study found that more experienced pharmacists reported a greater role in providing smoking cessation services than their less experienced counterparts [48]. It seems community pharmacists are more motivated and feel better prepared to provide pharmaceutical services over time. In contrast, Berenguer et al. found that older-generation pharmacists would lack the necessary clinical skills to manage drug therapy and are generally resistant to changing their practice [49].

Furthermore, pharmacists who attended continuing education on smoking cessation were more likely to engage in smoking cessation than pharmacists who had not participated in continuing education. In Wisconsin, USA, a study found that pharmacists participating in a tobacco cessation continuing professional program increased their self-efficacy and knowledge in patient counseling [50]. Additionally, the chances of pharmacists providing cessation counseling were higher for those who attended a smoking education program or were certified in smoking cessation counseling [51].

Lastly, we found that pharmacists interested in smoking cessation services engaged more in such services than those who did not. In line with those findings, Landau et al. found that interested pharmacists were comfortable providing hormonal contraception services (such as measuring blood pressure and weight, assessing medical history and risk, counseling patients on method use, and scheduling follow-up visits) [52]. Song and her colleagues reported that interest was a strong positive predictor of motivation to engage in specific content or activities [53].

The community pharmacies who participated in this study considered themselves to be reasonably confident when providing certain smoking cessation services, such as giving brief advice to help patients quit and discussing patient readiness to change smoking behavior with an overall mean confidence score of 2.7 (67.5%) (1 = not confident and 4 = highly confident). These findings are similar to those of surveys of community pharmacists conducted in Canada, Australia, and Qatar [36,37,38,54]. Saudi pharmacists, however, were less confident when addressing smoking issues unrelated to pharmacy visits. It has therefore been reported that students’ smoking cessation training has increased their perceived competence and confidence in delivering smoking cessation services [30].

The study found that community pharmacists encountered significant and common barriers to smoking cessation counseling. For example, 86% or more indicated that the most significant barriers to smoking cessation counseling were an inability to follow up and a lack of community pharmacists that included follow-up to motivate smokers to quit smoking. As part of counseling space, a lack of educational materials and a lack of time. First, being unable to follow up after providing the service was the most common issue considered to be a barrier. The 5 A’s (Ask, Advise, Assess, Assist, and Arrange) require pharmacists to arrange follow-ups. However, pharmacists may have thought arranging follow-ups would be problematic because patients rarely show interest [39]. Therefore, it was essential to have professional communication with community pharmacists that included follow-up to motivate smokers to quit smoking. As part of this protocol, pharmacists were instructed on how to conduct follow-ups and their purpose.

The lack of a private place to offer smoking cessation advice and counseling was also perceived as a barrier, which a Malaysian community pharmacist in an asthma management study also identified [55]. A study found that the lack of privacy in Saudi Arabian community pharmacies is a barrier preventing pharmacists from assuming a more active role in client counseling [56].

This study also identified the lack of suitable educational materials hindered service provision, and it is consistent with a related Thailand stud [57]. Because the Saudi Ministry of Health (MOH) provides materials for aiding in smoking cessation free of charge, health practitioners and individuals may receive a variety of educational materials for smoking cessation, including posters, stickers, and videos, from its MOH website. (https://www.moh.gov.sa, accessed on 2 May 2023). However, pharmacists may not be aware of this website; thus, outreach and communication to community pharmacists need to be enhanced.

The lack of time was cited as one of the significant barriers to providing smoking cessation services, similar to other studies [24,29,57]. Pharmacists will have more time to focus on patient-oriented activities if they are less involved in medication dispensing [58]. Hence, for pharmacists to adequately offer smoking cessation services to patients without causing significant disruption to their daily work routines, previous research suggests hiring more pharmacy technicians and better distinguishing the pharmacy technician role from the pharmacist role [38].

Community pharmacists are generally motivated and willing to provide smoking cessation services, with significant public health benefits to be gained. Tobacco smokers would have a higher chance of receiving advice and assistance from a pharmacist who is skilled and prepared to address smoking-related issues. Increasing participation in smoking cessation activities requires addressing the major barriers identified in this study. First, academic education and training are key strategies to increase confidence and skills among pharmacists. Only 19.2% of Saudi pharmacists indicated that they had attended training and education on smoking cessation; therefore, addressing this deficit is crucial. Education and training of pharmacists improve their knowledge of smoking cessation, confidence in delivering tobacco-related activities, and motivation to appropriately counsel patients [59,60,61]. A second area for improvement is the development of professional guidelines and protocols to guide and support community pharmacists in the provision of smoking cessation services. Developing accessible, evidence-based guidelines may enhance community pharmacists’ self-confidence in their services and improve the effectiveness of smoking cessation activities in the pharmacy sector [62]. To adequately support health organizations in implementing guidelines, interventions such as educational outreach and reminder systems may be used in health promotion. The 5As could be merged into smoking cessation guidelines, as they are the gold standard in the U.S., England, and other countries [37]. This study represented several limitations. First, the response rate was lower than optimal, so generalizing the findings should be done with caution. Second, A cross-sectional design was used in this study. As a result, the findings are considered correlational rather than causal. To assess change over time, a longitudinal study is needed.

Third, pharmacists were asked to self-report the information. Consequently, inaccurate responses may result from intentionally deceptive statements, poor recall of information, or misunderstandings of the question. Even so, the results remain valid in illustrating the role of community pharmacists in tobacco control and smoking cessation and highlighting a need for the pharmacy professions to extend to smoking cessation services. Another shortcoming was the absence of socio-demographic information such as marital and health status. Knowing such information may influence the interpretation of the responses to the survey.

Moreover, data were collected online using self-reported questionnaires based on the authors’ networks. Due to this, most respondents came from the Western region, where all the authors reside. Furthermore, the response may also be biased by a desire to respond in a socially desirable way. Finally, one survey cannot capture what community pharmacists are doing in Saudi Arabia regarding smoking cessation. Therefore, it may be necessary to conduct additional qualitative and observational studies to complement our quantitative findings. Despite its limitations, this study may provide legislators with the information they need to consider community pharmacy smoking cessation services as part of an official national health program. A nationwide community pharmacy setting could motivate smokers to seek smoking cessation services at their local pharmacy.

## 5. Conclusions

Community pharmacists in Saudi Arabia are generally willing to help the patient quit smoking. However, the pharmacists did not fully use the 5As approach to Tobacco cessation services. Unable to follow up with patients, lack of counseling space, lack of educational materials, and lack of time were cited as barriers to engaging in smoking cessation efforts should focus on increasing pharmacists’ confidence and training as well as overcoming all perceived obstacles since pharmacists are uniquely qualified to assist with smoking cessation.

## Figures and Tables

**Table 1 healthcare-11-01841-t001:** Demographics and Practice Characteristics of Community Pharmacists.

Characteristics	Respondents, n (%)
Age	
18–25	26 (7.0)
26–35	254 (68.6)
36–45	83 (22.4)
46–55	6 (1.6)
56–65	1(0.3)
Gender	
Male	333 (90.0)
Female	37 (10.0)
Region of practice	
Central region	49 (13.2)
Western region	249 (67.3)
Eastern region	27 (7.2)
Southern region	28 (7.5)
Northern region	17 (4.5)
Education level	
BSc	328 (88.6)
PharmD	32 (8.6)
MSc	8 (2.2)
PhD	2 (0.5)
Source of education	
Saudi university	79 (21.4)
Foreign university	391 (78.6)
Years practicing as a licensed pharmacist	
<5 years	112 (30.3)
5–10 years	129 (34.9)
>10 years	129 (34.9)
Work status	
Employed on a permeant basis	237 (73.8)
Employed as a casual	95 (25.7)
Type of community	
City	286 (77.3)
Town	52 (14.1)
Village	10 (2.7)
I don’t know	22 (5.9)
Smoking status	
Current smoker	53 (14.2)
Former smoker	21 (5.7)
Non-smoker	296 (80.0)
Attended an education or training program on smoking cessation	
Yes	71 (19.2)
No	299 (80.8)
Interesting in providing smoking cessation services	
Yes	251 (75.9)
No	89 (24.1)
The average number of cigarette smokers interacted with per week	
<5	189 (51.1)
5–10	62 (16.8)
11–15	37 (10.0)
>15	82 (22.2)
The average length of consultation per cigarette smoker	
0–5 min	263 (71.1)
6–10 min	87 (23.5)
11–15 min	20 (5.4)

**Table 2 healthcare-11-01841-t002:** Smoking Cessation-Related Activities.

Statement	Never n (%)	Rarely n (%)	Sometimes n (%)	Most of the Time n (%)	Always n (%)	Mean ± SD
Record						
Asking patients about their smoking cessation	134 (36.2)	108 (29.1)	70 (18.9)	30 (8.1)	28 (7.5)	2.3 ± 0.2
Recording the smoking status of your customers on a file	144 (38.9)	105 (28.4)	69 (18.6)	27 (7.3)	25 (6.8)	2.2 ± 1.2
Advise						
Providing brief advice to quit	24 (6.5)	113 (30.5)	81 (21.9)	74 (20.0)	78 (21.1)	3.2 ± 1.3
Linking advice to the presenting problem	26 (7.0)	104 (30.5)	96 (25.9)	78 (21.1)	66 (17.8)	3.2 ± 1.2
Discussing the effects of smoking on other family members	38 (10.3)	89 (24.1)	106 (28.6)	61 (16.5)	76 (20.5)	3.1 ± 1.3
Assess						
Assessing interest in quitting	29 (7.8)	109 (29.5)	99 (26.8)	73 (19.7)	60 (16.2)	3.1 ± 1.2
Assessing the level of nicotine dependence	44 (11.9)	118 (31.9)	93 (25.1)	64 (17.3)	51 (13.8)	3.0 ± 1.2
Assist						
Setting a quit date	69 (18.6)	126 (34.1)	89 (24.1)	52 (14.1)	33 (8.9)	2.6 ± 1.2
Developing a cessation plan	64 (17.3)	118 (31.9)	89 (24.1)	57 (15.4)	42 (11.4)	2.7 ± 1.2
Recommending nicotine replacement therapies such as gums and patches	53 (14.3)	83 (22.4)	105 (28.4)	82 (22.2)	47 (12.7)	3.0 ± 1.2
Providing quit materials	57 (15.4)	114 (30.8)	84 (22.7)	70 (18.9)	45 (12.2)	2.8 ± 1.3
Arrange						
Referring to a quitline program	58 (15.7)	110 (29.7)	97 (26.2)	64 (17.3)	41 (11.1)	2.8 ± 1.2
Referring to other pharmacists or general practitioners	57 (15.4)	126 (34.1)	95 (25.7)	53 (14.3)	39 (10.5)	2.7 ± 1.1
Follow-Up						
Following up on progress in giving up	66 (17.8)	119 (32.2)	91 (24.6)	52 (14.1)	41 (11.1)	2.7 ± 1.2

**Table 3 healthcare-11-01841-t003:** Relationship between Demographic and Practice Characteristics and Smoking Cessation-Related Activities.

Variables	Groups	N	Mean ± SD	*p*-Value
Age	18–25	26	2.64 ± 1.18	0.08
	26–35	254	2.75± 0.97	
	36–45	83	3.09 ± 1.02	
	46–55	6	98 ± 0.73	
	56–65	1	2.5 ± ----	
Gender	Male	333	2.86 ± 98	0.02 *
	Female	37	2.48 ± 1.13	
Region of practice	Central region	49	3.12 ± 0.80	0.37
	Western region	249	2.86 ± 1.01	
	Eastern region	27	2.85 ± 0.79	
	Southern region	28	2.45 ± 0.95	
	Northern region	17	2.37 ± 1.1	
Source of education	Saudi university	79	2.36 ± 0.82	0.01 *
	Foreign university	291	2.82 ± 0.76	
Type of employee	Permanent	273	2.72 ± 0.87	0.58
	Casual	95	2.70 ± 0.86	
Years practicing as a licensed pharmacist	<5 years	112	2.53 ± 1.02	0.01 *
	5–10 years	129	2.90 ± 0.95	
	>10 years	129	2.98 ± 0.99	
Type of community	City	286	2.81 ± 1.02	0.60
	Town	52	2.98 ± 0.95	
	Village	10	2.35 ± 0.99	
	I don’t know	22	2.80 ± 1.22	
Smoking status	Current smoker	53	2.47 ± 0.91	0.90
	Former smoker	21	2.63 ± 0.68	
	Non-smoker	296	2.88 ± 1.21	
Attended an education or training program on smoking cessation	Yes	71	3.0 ± 0.8	0.03 *
	No	299	2.7 ± 0.7	
Interesting in providing smoking cessation activities	Yes	281	2.9 ± 0.7	0.01 *
	No	89	2.5 ± 0.8	
The average number of cigarette smokers interacted with per week	<5	189	2.7 ± 1.0	0.11
	5–10	62	3.0 ± 0.9	
	11–15	37	2.9 ± 1.0	
	>15	82	2.9 ± 1.0	
The average length of consultation per cigarette smoker	0–5 min	263	2.7 ± 1.0	0.07
	6–10 min	87	3.9 ± 0.9	
	11–15 min	20	3.1 ± 0.9	

* *p*-value significant at <0.05.

**Table 4 healthcare-11-01841-t004:** Mean Rating of Importance of Confidence, System, and Practitioner Barriers.

Factor	Mean ± SD
System barriers (4 = High impact)	
No plan to implement protocols/guidelines	2.8 ± 1.2
Lack of smoking cessation protocols/guidelines	2.8 ± 1.2
Lack of knowledge of other support services	2.8 ± 1.23
Unable to follow up	2.9 ± 1.2
Lack of time	2.9 ± 1.2
Lack of educational materials	2.9 ± 1.9
Lack of counseling space	2.9 ± 1.2
Practitioner barriers (4 = High impact)	
Lack of necessary skills to assist patients to quit	2.8 ±1.2
The confidence I can be effective in helping to quit	2.8 ± 1.2
Patients might be alienated	2.9± 1.1
Not a worthwhile use of my time	2.6 ± 1.2
Confidence (5 = extremely confident)	
Raise smoking issues when they are related to the pharmacy visit	2.7 ± 0.9
Use brief advice to help people quit	2.9 ± 0.9
Discuss patient readiness to change smoking behavior	2.8 ± 0.9
Raise smoking issues when they are not related to the pharmacy visit	2.4 ± 0.9
Increase patient motivation to quit using specific counseling strategies	2.8 ± 0.9
Spend time assisting patients to quit	2.7 ± 0.9
Assess and refer to the pharmacist or GP	2.6 ± 0.9
Engage all staff members in a process	2.5 ± 0.9

Note: All Items were measured using a 5-point Likert scale ranging from 1 (“not confident”/no impact) to 5 (extremely confident/high impact). Abbreviation. SD, Standard Deviation.

## Data Availability

Data are available by contacting the corresponding author upon reasonable request.

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
