# Peer review of "Community Pharmacies’ Promotion of Smoking Cessation Support Services in Saudi Arabia: Examining Current Practice and Barriers"

_healthcare, 2023, doi:10.3390/healthcare11131841_

Round 1

Reviewer 1 Report

Title: Community Pharmacies’ Promotion of Smoking Cessation Support Services in Saudi Arabia: Examining Current Practice and Barriers

Thank you for providing a chance to review this manuscript.

Abstract:

·      Methods: Please include a brief description of the methods.

Introduction

·      Explain abbreviations when first mentioned in the text. Please review the complete text to attend to this recommendation. See page 1, line 34: “U.S.”; line 42 “COPD.”

·      Remove the author’s name from the text. Please review the complete document to attend to this recommendation. See page 1, line 41; page 2, line 70.

·      Please introduce the study's relevance in the first paragraph, and elaborate on the relevance of the community pharmacist.

Materials and methods

·      Include the ethical approval of the study.

·      Include the inclusion and exclusion criteria.

·      Measures:

o   In general, please provide a description of the measures with examples of the items for each of them. Also include the reliability of the measures included.

o   How was the questionnaire reviewed? Did they use specific steps or criteria for the review? If yes, please include a brief description.

Results

·      Results usually start with the description of the participants. Please review how the results are organized.

·      Please relocate tables to be part of the text and use three-line tables consistently.

·      Include 0 to those % values in the tables without decimals.

·      Be consistent in how many decimals are included.

·      Please include significance levels for the results in the text.

Discussion

·      Review the text for minor grammar errors.

·      Include numbers in the text.

·      The discussion could be improved by adding potential interventions to address the identified barriers.

Reviewer 2 Report

The manuscript (ID: healthcare-2411657) aimed to assess the role of community pharmacists in supporting their patients to quit smoking, as well as summarized their level of activities, confidence, and perceived barriers in Saudi Arabia. Correction was needed (major revision):

  • Lines 21-29: In the Abstract:
    • Explain and correct the sentence `Methods: briefly describe the main methods or treatments applied;`.
    • In the Abstract, state the study design that was applied in this study.
    • In the text of the Abstract, provide numerical data for the statistical significance of individual variables, in accordance with the corresponding data in the section Results in this manuscript..   
  • Lines 36-38: The very old references are reference No 2 and reference No 3. Check if those 2 references support the claim stated in this sentence. Cite references that present the calculation of the estimates for the year 2030.   
  • Line 56: State whether Saudi Arabia has signed and/or implemented the WHO Framework Convention on Tobacco Control.
  • Line 84: Indicate whether the above products are sold in pharmacies in Saudi Arabia (as in Italy, per reference No 31). In addition, state whether e-cigarettes are sold in pharmacies in Saudi Arabia.
  • Lines 160-168: Specify which questionnaire the stated results refer to.
  • Lines 206-232: In the text, provide numerical data for the statistical significance of individual variables, in accordance with the corresponding table.
  • Lines 385-399: Supplement the paragraph on the limitations of this study and a discussion on whether a potential source for the limitations of the study was the applied study design (a cross-sectional study design), the lack of other socio-demographic and other characteristics of the respondents (marital status, children, state of health, etc). In particular, the following should be discussed as limitations:
    • The result that 51.1% of respondents gave the information that `The average number of cigarette smokers interacted with per week' was `< 5'. So, in this study there were respondents who did not even interact with patients who want to quit smoking. Discuss.   

The quality English language is appropriate.  

Round 2

Reviewer 2 Report

Thank you for the opportunity to review the revised version of the manuscript ID: healthcare-2411657. Overall, the corrections made have significantly improved this manuscript. As for my comments, the authors answered all questions (in all chapters, from Abstract, ..., to References), and provided appropriate explanations. I thank the authors.    

The quality of English language is appropriate.